# Inter- and Intra-Molecular Organocatalysis of S_N_2 Fluorination by Crown Ether: Kinetics and Quantum Chemical Analysis

**DOI:** 10.3390/molecules26102947

**Published:** 2021-05-15

**Authors:** Young-Ho Oh, Wonhyuck Yun, Chul-Hee Kim, Sung-Woo Jang, Sung-Sik Lee, Sungyul Lee, Dong-Wook Kim

**Affiliations:** 1Department of Applied Chemistry, Kyung Hee University, Duckyoung-daero 1732, Yongin City 446-701, Korea; chem_yhoh@daum.net (Y.-H.O.); dughtndk23@naver.com (S.-W.J.); int_shik@hotmail.com (S.-S.L.); 2Department of Chemistry, Inha University, 100 Inha-ro, Nam-gu, Incheon 402-751, Korea; wh0608777@gmail.com (W.Y.); cg7633@naver.com (C.-H.K.)

**Keywords:** organocatalysis, S_N_2 fluorination, crown ether, quantum chemistry

## Abstract

We present the intra- and inter-molecular organocatalysis of S_N_2 fluorination using CsF by crown ether to estimate the efficacy of the promoter and to elucidate the reaction mechanism. The yields of intramolecular S_N_2 fluorination of the veratrole substrates are measured to be very small (<1% in 12 h) in the absence of crown ether promoters, whereas the S_N_2 fluorination of the substrate possessing a crown ether unit proceeds to near completion (~99%) in 12 h. We also studied the efficacy of intermolecular rate acceleration by an independent promoter 18-crown-6 for comparison. We find that the fluorinating yield of a veratrole substrate (leaving group = −OMs) in the presence of 18-crown-6 follows the almost identical kinetic course as that of intramolecular S_N_2 fluorination, indicating the mechanistic similarity of intra- and inter-molecular organocatalysis of the crown ether for S_N_2 fluorination. The calculated relative Gibbs free energies of activation for these reactions, in which the crown ether units act as Lewis base promoters for S_N_2 fluorination, are in excellent agreement with the experimentally measured yields of fluorination. The role of the metal salt CsF is briefly discussed in terms of whether it reacts as a contact ion pair or as a “free” nucleophile F^−^.

## 1. Introduction

S_N_2 reactions [1,2,3,4,5,6,7,8,9,10,11,12,13] are such fundamental chemical transformations that have been studied for so long that the mechanism of this important and useful process is usually considered to be perfectly understood. There have been, however, some studies [4,5,8,13,14,15] proposing that the mechanism of S_N_2 reactions may not be so unambiguous. Complications may arise from the influence of counter-cation [16] and/or solvent [15], which have usually been omitted in the discussions for S_N_2 reactions. It was commonly assumed that the counter-cation is sufficiently far from the nucleophile, so that the counter-cation/nucleophile may be considered to be a solvent-separated ion-pair. In this regard, some useful wisdom has been frequently invoked concerning the role of counter-cation and solvent for S_N_2 reactions: alkali metal cations are considered to be inefficient as counter-cation, and protic solvents (water, alcohols) are to be avoided because of the harmful hydrogen bonding through –OH with the nucleophile. In a series of investigations, however, we have demonstrated [17,18,19,20,21,22,23] that these well-known concepts are not always true, showing that using alkali metal salts (such as CsF or KBr) in combination with Lewis base promoter/solvent (bulky alcohols, oligoethylene glycols, ionic liquids and derivatives et al.) may lead to excellent S_N_2 reactions without many side-products (usually with >90% S_N_2 yield within several hours).

Here, we present the S_N_2 fluorination promoted by crown ether [23,24,25,26,27,28] using the alkali metal salt CsF to estimate the efficacy of the promoter and to elucidate the mechanism^-^. We employ different types of substrates (**1** vs. **2**) for comparison. The *intramolecular* crown ether **1** and the veratrole-Oms substrate are chosen to isolate the effects of substrate structures on the Gibbs free energy of activation *G*^‡^ from other constituents (such as the collision frequency, orientation of collision, etc.) in the pre-exponential factor in Arrhenius equation for rate constants. In the S_N_2 fluorination of crown ether we find that fused reactant 1 proceeds much faster (~99% in 12 h) than that of 2, clearly demonstrating the efficacy of the intramolecular promoter.

Moreover, we observe that the yield of intramolecular S_N_2 fluorination of substrate **2** (Veratrole-OMs) almost exactly follows the kinetics of **1** in the presence of 18-crown-6 **3** to near completion in 12 h, indicating that very similar mechanisms are involved both in intra- and inter-molecular organocatalysis of the crown ether promoter for S_N_2 fluorination. We examine the mechanisms of fluorination for the reactants to estimate the role of intramolecular crown ether unit in **1** and the two methoxy groups in **2** for enhancing the S_N_2 rates. Organocatalysis of S_N_2 fluorination of 2 by an independent (intramolecular) crown ether (18-crown-6), which is a more conventional type of reaction promoted by crown ether, is also studied. We show that the calculated relative Gibbs free energies of activation for these reactions are in excellent agreement with the experimentally measured yields of fluorination. Our calculations also illustrate that the mechanisms of these observed kinetics are such that the crown ether units act as Lewis base promoters [17,18,19,20,21,22,29] for S_N_2 fluorination. Brief discussion is given concerning the role of the metal salt CsF in relation to whether it reacts as a contact ion pair [30] or not.

## 2. Materials and Methods

General remarks. Unless otherwise noted, all reagents and solvents were commercially available. TLC analysis was performed using Merck silica gel 60F_254_ plates. Visualization on TLC was monitored by UV light (254 nm). Flash chromatography was performed with 230–400 mesh silica gel. ^1^H and ^13^C NMR spectra were recorded on a 400 MHz spectrometer (Bruker, Billerica, USA), and chemical shifts were reported in δ units (ppm) relative to tetramethylsilane. High resolution mass spectra were obtained at the Korea Basic Science Institute (Daegu, Korea).

Typical Procedure for S_N_2 Fluorination in Figure 1**.** CsF (46 mg, 0.3 mmol) was added to the mixture of substrates **1**–**3** (0.1 mmol) in CH_3_CN (1.0 mL). The reaction mixture was stirred at 80 °C. At each time point, 0.1 mL of mixture was sampled. The reaction mixture was cooled to room temperature and the solvent (CH_3_CN) was removed under vacuum to concentrate the product. Ratios of compounds in reaction mixtures were determined using ^1^H NMR spectroscopy.

4-Fluoropropylbenzo-18-crown 6-ether (**1a**). According to the typical procedure for fluorination, **1a** (37 mg, 99%) was obtained as a yellow oil after flash column chromatography (10% MeOH/MC); ^1^H NMR (400 MHz, CDCl_3_) δ 1.90–2.03 (m, 2H), 2.66 (t, *J* = 7.7 Hz, 2H), 3.68–3.77 (m, 12H), 3.90–3.93 (m, 4H), 4.12–4.16 (m, 4H), 4.45 (dt, *J* = 47.3, 6.0 Hz, 2H), 6.70–6.72 (m, 2H), 6.81 (d, *J* = 8.0 Hz, 1H); ^13^C NMR(100 MHz, CDCl_3_) δ 30.94 (d, *J* = 4.8 Hz), 32.27 (d, *J* = 19.3 Hz), 69.32, 69.49, 69.88, 69.90, 70.90, 70.94, 83.21 (d, *J* = 164.7 Hz), 114.74, 115.00, 121.22, 134.55, 147.47, 149.15; MS (FAB) 372 (M^+^), 149 (100); HRMS (FAB) calculated for C_19_H_29_FO_6_ (M^+^) 372.1948, found 372.1950.

4-(3-Fluoropropyl)-1,2-dimethoxybenzene (**2a**). Prepared according to the typical procedure for fluorination except the use of 18-crown-6 (26 mg, 0.1 mmol), **2a** (19 mg, 98%) was obtained as a colorless oil after flash column chromatography (20% EtOAc/hexane); ^1^H NMR (400 MHz, CDCl_3_) δ 1.92–2.05 (m, 2H), 2.69 (t, *J* = 7.7 Hz, 2H), 3.85 (s, 3H), 3.87 (s, 3H), 4.47 (dt, *J* = 47.1, 6.0 Hz, 2H), 6.72–6.74 (m, 2H), 6.80 (d, *J* = 8.2 Hz, 1H); ^13^C NMR(100 MHz, CDCl_3_) δ 30.97 (d, *J* = 4.8 Hz), 32.29 (d, *J* = 19.3 Hz), 55.88, 55.98, 83.15 (d, *J* = 163.8 Hz), 111.39, 111.90, 120.37, 133.78, 147.42, 148.98; MS (EI) 198 (M^+^), 151 (100); HRMS (EI) calculated for C_11_H_15_FO_2_ (M^+^) 198.1056, found 198.1057.

## 3. Computational Details

All calculations were carried out by M06-2X [31] density functional theory method with 6-311G(d,p) [32,33] basis set for C, H, O, S, F atoms and LANL2DZ basis set for Cs atom with their corresponding effective core potential [34] as implemented in G09 software package [35]. The SMD model [36] was used to consider the effects of solvation by acetonitrile. All frequencies for pre-/post- reaction complexes were confirmed to be real and one of those for transition state was ascertained to be imaginary. To determine the unique reaction pathway, the intrinsic reaction coordinate analysis was applied.

## 4. Results and Discussion

Figure 1 depicts our present scheme designed to verify the effects of crown ethers on S_N_2 fluorination. Substrate **1** is a crown ether unit fused with a phenyl ring at which S_N_2 reaction occurs with the mesylate (-OMs) as the leaving group. We introduce the veratrole substrate **2** with -OMs as the leaving group, in which the crown ether moiety is absent, for comparison. The two methoxy groups in **2** are to simulate the promoting effects caused by the crown ether unit of **1**. First, we carried out intramolecular S_N_2 fluorination of **1** promoted by the fused crown ether unit, in comparison with the S_N_2 fluorination of **2**. Second, we observed the S_N_2 kinetics for fluorination of **2** in the presence of intermolecular promotion by a separate crown ether **3** to compare with intramolecular rate acceleration by the fused crown ether unit in **1**.

The S_N_2 fluorination yields of **1** and **2** substrates are presented in Figure 1 as functions of time. We observe several intriguing kinetic features: first, the intramolecular S_N_2 fluorination was observed to be nearly complete within 12 h, demonstrating the efficacy of the intramolecular promoter of crown ether, whereas the same reaction of **2** hardly proceeds (~1% in 24 h). Thus, S_N_2 fluorination proceeds extremely slowly in the absence of the promoting effects of the crown ether moiety, indicating that the two methoxy groups in the veratrole substrate **2** are far less efficient than the crown ether ring for promoting the S_N_2 fluorination using CsF. These differences in the reactivity of the substrates **1** and **2** are well accounted for by the Gibbs energies of activation (*G*^‡^) for the two cases, as described below. Second, a more intriguing observation depicted in Figure 1 is that the yield of *intermolecular* fluorination of **2** in the presence of the separate promoter **3** (18-crown-6) follows an almost indistinguishable time evolution of S_N_2 *intramolecular* fluorination of **1**: Both reactions arrive at ~80% in 6 h, and at near completion within 12 h. It seems that very similar mechanisms are involved both in intramolecular 18-crown-6 promoted S_N_2 fluorination of **1** and in intermolecular 18-crown-6 promoted fluorination of **2** by **3**, probably via the structurally similar pre-reaction complexes and transition states (TSs).

The calculated energetics for S_N_2 fluorination reactions of **1** and **2** are depicted in Figure 2. The Gibbs free energy of activation of S_N_2 fluorination of **1** and **2** are calculated to be 6.8 and 14.3 kcal/mol, respectively. This clearly accounts for the observed poor yields of fluorination for **2** relative to that of the intramolecular fluorination of **1** by the much higher Gibbs free energy of activation of S_N_2 fluorination of **2** than that of **1**. The TSs in S_N_2 fluorination of **1** are depicted in Figure 3 (NMR spectra and Cartesian coordinates of the pre- and post-reaction complexes and the TSs are given in Appendix A). In S_N_2 fluorination of **1**, the nucleophile F^−^ seems to react with **1** either in close contact with (*G*^‡^ = 6.85, *R*_Cs-F_ = 3.238 Å, Sub1-Ι)**,** or far away (as compared with *R*_Cs-F_ = 2.836 Å of CsF in CH_3_CN) from the counter-cation Cs^+^ (*G*^‡^ = 6.82 kcal/mol, *R*_Cs-F_ = 7.724 Å, Sub1-ΙΙ) in the TSs, both of which are predicted to contribute almost equally because of their very similar Gibbs free energies of activation. For fluorination of the substrates **2**, the distance between Cs^+^ and F^−^ in the TS is very small (*R*_Cs-F_ = 2.992 Å).

Figure 4 depicts the calculated energetics and structure of the TS for S_N_2 fluorination reactions of **2** in the presence of **3** (18-crown-6). It seems that the latter structure formed between **2** and **3** for intermolecular S_N_2 fluorination of **2** strongly resembles the TS for intramolecular S_N_2 fluorination of **1** shown in Figure 3. From this, the role of 18-crown-6 is clear: the metal salt Cs^+^F^−^ coordinates to **3**, just as in S_N_2 fluorination of **1**, and the O atoms in 18-crown-6 are acting as a Lewis base alleviating the Coulombic influence of the counter-cation Cs^+^ to enhance the nucleophilicity of F^−^, as previously elucidated for other promoters [17,18,19,20,21,22,23] (bulky alcohols, oligoethylene glycols, and ionic liquids).

The calculated Gibbs free energy of activation (6.2 kcal/mol) for fluorination of **2** in the presence of 18-crown-6 shown in Figure 4a is very close to, but slightly lower than that (6.8 kcal/mol) of **1** in Figure 2. The slightly larger observed yield for fluorination of **1** presented in Figure 1 may indicate that a small portion of **2** and **3** does not form 1:1 complex. It may also be the results of the fact that the intramolecular case (Figure 2) is more exothermic than the intermolecular case (Figure 4) (Gibbs free energy of reaction −12.59 vs. −8.38 kcal/mol). It is also noteworthy that in this latter process CsF reacts as a contact ion pair with *R*_Cs-F_ = 3.072 Å in the TS.

## 5. Conclusions

In this study, we have demonstrated the excellent efficacy of crown ethers for the rate enhancement of both intramolecular and intermolecular S_N_2 fluorination using CsF. Interestingly, similar mechanisms were shown to work for both processes with comparable kinetics. The role of the crown ethers as Lewis base promoters interacting with the counter cation Cs^+^ to reduce its strong Coulombic influence on the nucleophile F^−^ is in line with our previous works on the organocatalysis of S_N_2 reactions. One critical observation made here is that the crown ethers do not seem to separate counter-cation Cs^+^ from the nucleophile in some cases, and that the metal salt CsF may act as an ion pair, in contrast to the conventional concept of crown ether. This intriguing topic will be discussed in more detail in our subsequent works.

## Figures and Tables

**Figure 1 molecules-26-02947-f001:**
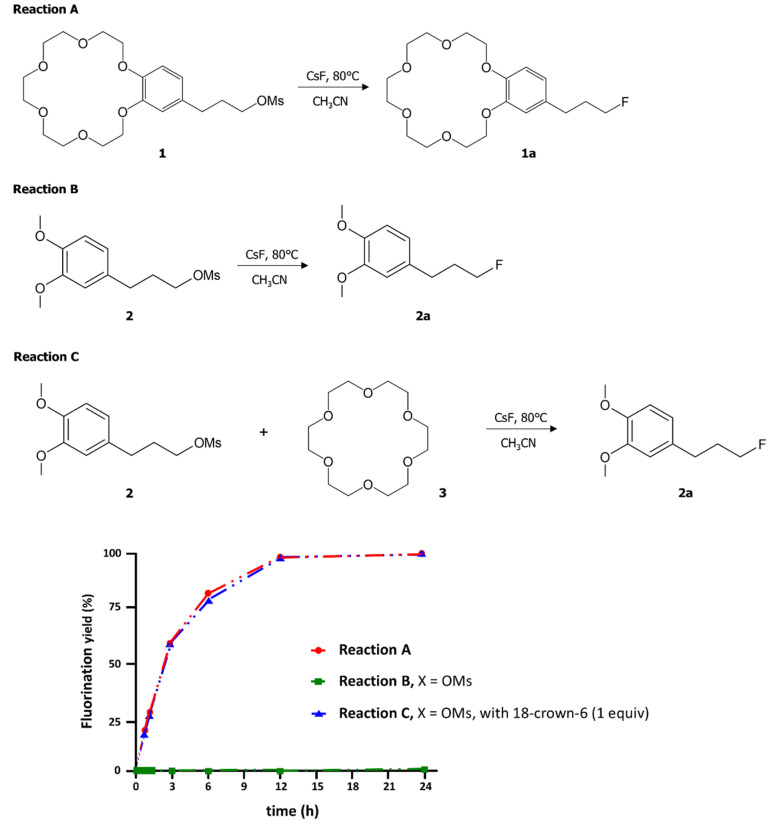
Intra- and inter-molecular 18-crown-6 promoted nucleophilic fluorination in comparison with S_N_2 fluorination of the veratrole substrate **2**. (A) Reaction of intramolecular 18-crown-6 integrated mesylate **1**. (B) Reaction of mesylate **2** in absence of 18-crown-6. (C) Reaction of mesylate **2** with in the presence of **3**. Quantity of the products was determined using ^1^H NMR spectroscopy.

**Figure 2 molecules-26-02947-f002:**
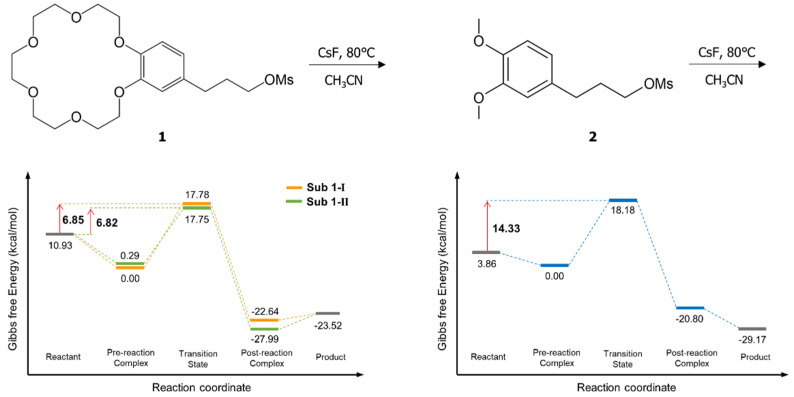
Energetics of S_N_2 fluorination of **1** and **2** using CsF. Gibbs free energy in kcal/mol.

**Figure 3 molecules-26-02947-f003:**
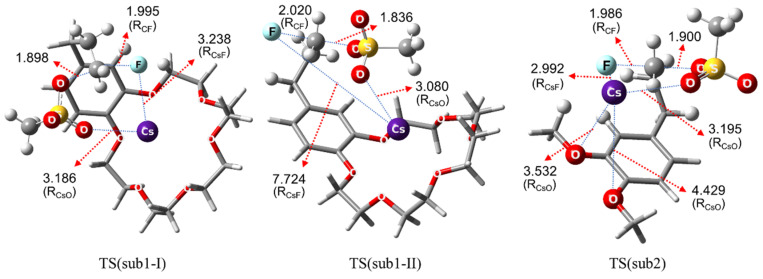
Transition states in S_N_2 fluorination of **1** and **2** using CsF. Bond lengths in Å.

**Figure 4 molecules-26-02947-f004:**
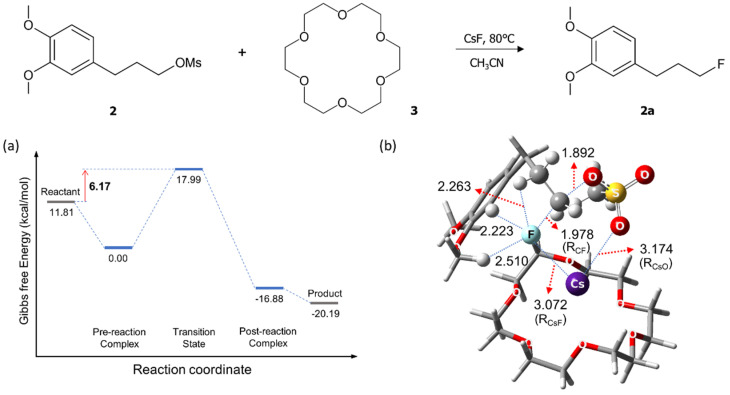
(**a**) Energetics and (**b**) structure of the transition state for S_N_2 fluorination of **2** in the presence of **3** (18-crown-6). Gibbs free energy in kcal/mol, and bond lengths in Å.

## Data Availability

Not applicable.

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
