# Peer review of "Inter- and Intra-Molecular Organocatalysis of SN2 Fluorination by Crown Ether: Kinetics and Quantum Chemical Analysis"

_molecules, 2021, doi:10.3390/molecules26102947_

Round 1
Reviewer 1 Report
This article presents the intra- and inter-molecular organocatalysis
of SN2 fluorination by crown ether. Both experimental and theoretical
results are presented. Ab initio calculations done by the authors nicely
correlate with the experimental kinetic results they had obtained.
While this article is definitely publishable, it needs to be revised
for the following reasons and the changes outlined below needs to be
done so that it would not confuse the audience.
1. Figure 1 caption including the labeling is slightly misleading for
the following reasons. In the top, Reaction A, B, and C are given.
In the bottom part of the figure, I do not see the Reaction C label. Both are
mentioned as B. However, I think the one with 18-crown-6 (intermolecular)
is, Reaction C (the blue color used for fluorination yield). This must be corrected.
2. In the introduction as well as in Section 4 (Results), the authors
mention that the 18-crown-6 is "4". However, label "3" is used for
Reaction C in Figure 1. This also must be corrected. Because, according
to authors in the text label 3 corresponds to Veratrole substrate with
-Br leaving group.
3. I also do not see the kinetics of substrate 3 in Figure 1. I mean the one with -Br leaving group. Only the kinetics of 1 and 2 are given. Are there any specific reasons for this?
4. The quality of the chemdraw images in Figure 1 is not the same as
the qulity of chemdraw Figures in 2 and 4. Figure 1 needs to be
improved apart from the above corrections mentioned.
5. Please change 6-311G** to 6-311G(d,p).
6. Line 101; transition states was --> transition state was
7. Line 121; fluorination of was --> fluorination was
8. Line 130; 18-crown-6-ether --> 18-crown-6. It is understood by
everyone that 18-crown-6 is a crown ether. Therefore, no need to
mention it as "18-crown-6-ether".
9. Please give the labels 1, 2 and 3 in the substrates too in Figure
2 chemdraw images for an easy understanding by the readers.
10. Line 143; for 1 --> of 1.
11. Figure 4; Label the chemdraw images on the left hand side.
Substrate as 2 and 18-crown-6 as 4.
Remarks: The authors could also explain the higher yield of fluorination
in the case of 1 by using the concept of reaction energy.
In the intramolecular case (figure 1; substrate 1), the reaction energy is
(Product-Reactant; -23.52+10.93=) -12.59 kcal/mol.
In the intermolecular case (figure 4; substrate 2 with 4), the reaction energy is -20.19+11.81 = -8.38 kcal/mol.
Therefore, the former is slightly more exothermic than the latter.
I appreciate the critical observation made by the authors that states that "CsF may act as an ion-pair, in contrast to conventional concept of crown ether".
Author Response
We thank the Reviewers for many helpful comments.
Reviewer 1:
1.Figure 1 caption including the labeling is slightly misleading for
the following reasons. In the top, Reaction A, B, and C are given.
In the bottom part of the figure, I do not see the Reaction C label. Both are
mentioned as B. However, I think the one with 18-crown-6 (intermolecular)
is, Reaction C (the blue color used for fluorination yield). This must be corrected.
Figure 1 and the caption are corrected accordingly.
- In the introduction as well as in Section 4 (Results), the authors
mention that the 18-crown-6 is "4". However, label "3" is used for
Reaction C in Figure 1. This also must be corrected. Because, according
to authors in the text label 3corresponds to Veratrole substrate with
-Br leaving group.
It is corrected.
- I also do not see the kinetics of substrate 3 in Figure 1. I mean the one with -Br leaving group. Only the kinetics of 1and 2are given. Are there any specific reasons for this?
It was our mistake. We apologize for the confusion. We correct in pp 2, line # 52:
veratrole-X (X = OMs, Br) à veratrole-OMs
- The quality of the chemdraw images in Figure 1 is not the same as
the qulity of chemdraw Figures in 2 and 4. Figure 1 needs to be
improved apart from the above corrections mentioned.
Figure 1 is revised.
- Please change 6-311G** to 6-311G(d,p):
It is revised to 6-311G(d,p)
- Line 101; transition states was --> transition state was:
It is corrected.
- Line 121; fluorination of was --> fluorination was:
It is corrected.
- Line 130; 18-crown-6-ether --> 18-crown-6. It is understood by
everyone that 18-crown-6 is a crown ether. Therefore, no need to
mention it as "18-crown-6-ether":
It is corrected.
- Please give the labels 1, 2and 3in the substrates too in Figure
2 chemdraw images for an easy understanding by the readers.
It is corrected.
- Line 143; for 1--> of 1.
It is corrected.
- Figure 4; Label the chemdraw images on the left hand side.
Substrate as 2and 18-crown-6 as 4.
It is corrected.
- The authors could also explain the higher yield of fluorination
in the case of 1by using the concept of reaction energy.
In the intramolecular case (figure 1; substrate 1), the reaction energy is
(Product-Reactant; -23.52+10.93=) -12.59 kcal/mol.
In the intermolecular case (figure 4; substrate 2 with 4), the reaction energy is -20.19+11.81 = -8.38 kcal/mol. Therefore, the former is slightly more exothermic than the latter.
We add the following paragraph at the end of the Results section:
It may also be the results of the fact that the the intramolecular case (Figure 1) is more exothermic than the intermolecular case (Figure 4) (Gibbs free energy of reaction -12.59 vs. -8.38 kcal/mol)

Reviewer 2 Report
Dear Authors and Editors,
Here are my major comments for the reviewed manuscript:
- The paragraph in the Introduction section starting with “Here we present…” is confusing as it contains the summarization of the results in the very beginning of the paper. It should be either replaced to Conclusions sections or reworked. However, the motivation for the choice of the objects of investigation is not evident from the text.
- The scheme resolution is rather low, it is difficult to read the legend.
- Actually, I could not reliably put into correspondence the text of experimental section, figure 1 and its caption. For example:
-“Second, we observed the SN2 kinetics for fluorination of 2 in the presence of intermolecular promotion by a separate crown ether 4” – in fig. 1, if I understand correctly, the separate crown is marked as “3”.
-In reaction C 2 does not contain X abbreviation, OMs is already written in the structure
-in the part of Fig.1 with fluorination yield we see in the legend: Reaction B with crown. But if I understand the research design correctly, it is Reaction C from the scheme above.
-Where is the definition of substrate 3 with –Br substituent?
-“The SN2 fluorination yields of 1, 2 and 3 substrates are presented in Figure 1 as functions of time.” – I do not see substrate 3 except for the separate crown on the scheme.
All in all, this part is rather misleading and inconsistent.
- What is the distance of CsF in MeCN solution in the absence of other molecules?
- What is the essential role of the usage of the separate crown instead, for example, of an efficient substrate 1? What practical tasks solve the usage of a combination of substrate and separate crown?
Other minor remarks:
- Abstract: abbreviation of the substrate numbers 1, 2 and 3 is not informative for the reader.
- Unnecessary abbreviations: Hrs, Calcd
- "solvents were removed" – the authors specify only MeCN as a solvent, what other solvents do they use?
Author Response
We thank the Reviewers for many helpful comments.
Reviewer 2:
- The paragraph in the Introduction section starting with “Here we present…” is confusing as it contains the summarization of the results in the very beginning of the paper. It should be either replaced to Conclusions sections or reworked. However, the motivation for the choice of the objects of investigation is not evident from the text.
- We revise the paragraph in Introduction from:
We find that the yields of intramolecular SN2 fluorination of 2 and 3 (Veratrole-X, X = -OMs, -Br) are measured to be extremely small (< 1 % in 12 hrs). On the other hand, the SN2 fluorination of 1 proceeds to near completion (~99 %) in 12 hrs, illustrating the efficacy of intramolecular crown ether promoter for SN2 fluorination. Moreover, we observe that the yield of intramolecular SN2 fluorination of Substrate 2 (Veratrole-OMs) almost exactly follows the kinetics of 1 in the presence of 18-crown-6 4 to near completion in 12 hrs, indicating that very similar mechanisms are involved both in intra- and intermolecular organocatalysis of the crown ether promoter for SN2 fluorination.
To (pp. 2, line # 54 - 60):
We find that the SN2 fluorination of 1 under the influence crown ether proceeds much faster (~99 % in 12 hrs) than that of 2 and 3, clearly demonstrating the efficacy of the promoter. Moreover, we observe that the yield of intramolecular SN2 fluorination of Substrate 2 (Veratrole-OMs) almost exactly follows the kinetics of 1 in the presence of 18-crown-6 3 to near completion in 12 hrs, indicating that very similar mechanisms are involved both in intra- and intermolecular organocatalysis of the crown ether promoter for SN2 fluorination.
- The scheme resolution is rather low, it is difficult to read the legend.
The figure for the scheme is revised.
- Actually, I could not reliably put into correspondence the text of experimental section, figure 1 and its caption. For example:
-“Second, we observed the SN2 kinetics for fluorination of 2 in the presence of intermolecular promotion by a separate crown ether 4” – in fig. 1, if I understand correctly, the separate crown is marked as “3”.
It is corrected.
-In reaction C 2 does not contain X abbreviation, OMs is already written in the structure
It is corrected.
-in the part of Fig.1 with fluorination yield we see in the legend: Reaction B with crown. But if I understand the research design correctly, it is Reaction C from the scheme above.
It is corrected in Figure 1.
-Where is the definition of substrate 3 with –Br substituent?
It was our mistake. We apologize for the confusion. Vetratrole – Br is not included in our present work. We correct in pp 2, line # 52:
veratrole-X (X = OMs, Br) à veratrole-OMs
-“The SN2 fluorination yields of 1, 2 and 3 substrates are presented in Figure 1 as functions of time.” – I do not see substrate 3 except for the separate crown on the scheme.
Reaction C is SN2 fluorination of substrate 2 under the influence of 3 (18-crown-6). We revised Figure 1 and the caption accordingly. We apologize for the confusion.
- What is the distance of CsF in MeCN solution in the absence of other molecules?
- We add the paragraph (pp. 3, line # 132-133)
(as compared with RCs-F = 2.836 â„« of CsF in CH3CN)
- What is the essential role of the usage of the separate crown instead, for example, of an efficient substrate 1? What practical tasks solve the usage of a combination of substrate and separate crown?
- We add (pp. 3, line # 118-119)
, which is more conventional type of reaction promoted by crown ether,
Other minor remarks:
- Abstract: abbreviation of the substrate numbers 1, 2 and 3 is not informative for the reader.
- We revise the Abstract to:
We present the intra- and intermolecular organocatalysis of SN2 fluorination using CsF by crown ether to estimate the efficacy of the promoter and to elucidate the reaction mechanism. We employ three substrates 1, 2 and 3 for comparison. The yields of intramolecular SN2 fluorination of the veratrole substrates are measured to be very small (< 1 % in 12 hrs) in the absence of crown ether promoters, whereas the SN2 fluorination of the substrate possessing a crown ether unit proceeds to near completion (~99 %) in 12 hrs. We also studied the efficacy of intermolecular rate acceleration by an independent promoter 18-crown-6 for comparison. We find that the fluorinating yield of a veratrole substrate (leaving group = -OMs) in the presence of 18-crown-6 follows the almost identical kinetic course as that of intramolecular SN2 fluorination, indicating the mechanistic similarity of intra- and intermolecular organocatalysis of the crown ether for SN2 fluorination. The calculated relative Gibbs free energies of activation for these reactions, in which the crown ether units act as Lewis base promoters for SN2 fluorination, are in excellent agreement with the experimentally measured yields of fluorination.
- Unnecessary abbreviations: Hrs, Calcd
It is revised.
"solvents were removed" – the authors specify only MeCN as a solvent, what other solvents do they use?
It is revised to: the solvent (CH3CN) was removed
Round 2
Reviewer 2 Report
Dear Authors and Editors,
As I had written in the previous review round, there are still very serious inconsistencies in the text, the authors should treat the text more attentively.
-Although Abstract has been rewritten, there are still questions about it.
Number abbreviations in the abstract do not provide the readers essential information. Also, there are no "three substrates", actually there is only one substrate, in the limit case - two, but crown ether itself is not a substrate in any case.
-As I wrote before, the end of the Introduction section contains a summarization of the results performed in this work but not a deeper insight into the choice of the substrate, the role and mechanism of crown ether reaction promoting, and so on and so forth what the readers expect to see in the introduction section to get a better view on the results already obtained in the discussed research field.
Moreover, the newly written phrase from line 54 and further is misleading and ambiguous: "We find that the SN2 fluorination of 1 under the influence crown ether proceeds much faster (~99 % in 12 hrs) than that of 2 and 3, clearly demonstrating the efficacy of the promoter." According to kinetic curve, the fluorination of 1 is almost identical to 2+3 and the authors mention it themselves. Here the meaning of the phrase is incorrect as 2+3 is almost equal to 1, and 3 itself is not a substrate and can not undergo fluorination separately.
Such inconsistent style of writing really complicates the understanding of the work!
Besides, I mentioned in the previous review round about Br- substituent of the substrate. The authors claim that this was an error, but how can it be, that all the data, including NMR, transition state illustration are still present in the supplementary file?
I think the authors should carefully reread all the text and make it consistent and easy to read for the broad audience.
Line 134 "bo" is a typo or error
Author Response
We thank the Reviewer for many helpful comments.
- Number abbreviations in the abstract do not provide the readers essential information. Also, there are no "three substrates", actually there is only one substrate, in the limit case - two, but crown ether itself is not a substrate in any case.
The following sentence is deleted from the Abstract.
We employ three substrates 1, 2 and 3 for comparison.
We add at the end of the Abstract. The role of the metal salt CsF is briefly discussed, whether it reacts as a contact ion pair of as a “free” nucleophile F-.
- The end of the Introduction section contains a summarization of the results performed in this work but not a deeper insight into the choice of the substrate, the role and mechanism of crown ether reaction promoting, and so on and so forth what the readers expect to see in the introduction section to get a better view on the results already obtained in the discussed research field.
We add the following sentences:
To the Introduction (pp. 2, line # 61-65): We examine the mechanisms of fluorination for the reactants to estimate the role of intramolecular crown ether unit in 1 and the two methoxy groups in 2 for enhancing the SN2 rates. Organocatalysis of SN2 fluorination of 2 by an independent (intramolecular) crown ether (18-crown-6), which is more conventional type of reaction promoted by crown ether, is also studied.
At the end of the Introduction: Brief discussion is given concerning the role of the metal salt CsF, whether it reacts as a contact ion pair or not.
- The newly written phrase from line 54 and further is misleading and ambiguous: "We find that the SN2 fluorination of 1 under the influence crown ether proceeds much faster (~99 % in 12 hrs) than that of 2 and 3, clearly demonstrating the efficacy of the promoter." According to kinetic curve, the fluorination of 1 is almost identical to 2+3 and the authors mention it themselves. Here the meaning of the phrase is incorrect as 2+3 is almost equal to 1, and 3 itself is not a substrate and can not undergo fluorination separately.
We revise that part (pp. 2, line # 55-57) to:
We find that the SN2 fluorination of crown ether – fused reactant 1 proceeds much faster (~99 % in 12 hrs) than that of 2, clearly demonstrating the efficacy of the intramolecular promoter.
- I mentioned in the previous review round about Br- substituent of the substrate. The authors claim that this was an error, but how can it be, that all the data, including NMR, transition state illustration are still present in the supplementary file?
We deeply apologize for our mistake. We revise the supplementary material, deleting all data for veratrole-OMs.
- Line 134 "bo" is a typo or error
It is corrected to: both
- We add the following reference for CIP SN2 reaction:
- Laloo, J. Z. A.; Rhyman, L.; Ramasami, P.; Bickelhaupt, F. M.; Cózar, A. d. Ion-Pair SN2 Substitution: Activation Strain Analyses of Counter-Ion and Solvent Effects, Chem. Eur. J. 2016, 22, 4431-4439.